# Exploring the Biocultural Nexus of *Gastrodia elata* in Zhaotong: A Pathway to Ecological Conservation and Economic Growth

**DOI:** 10.3390/biology14070846

**Published:** 2025-07-11

**Authors:** Yanxiao Fan, Menghua Tian, Defen Hu, Yong Xiong

**Affiliations:** 1School of Ethnic Medicine, Yunnan Minzu University, Kunming 650504, China; fanyanxiao0510@163.com; 2Key Laboratory of Chemistry in Ethnic Medicinal Resources, Yunnan Minzu University, Ministry of Education, Kunming 650504, China; 3Zhaotong Institute of Gastrodia, Zhaotong 657000, China; tianmenghua@126.com (M.T.); 18087037757@163.com (D.H.); 4Yunnan Key Laboratory of Gastrodia and Fungi Symbiotic Biology, Zhaotong University, Zhaotong 657000, China; 5Yunnan Engineering Research Center of Green Planting and Processing of Gastrodia, Zhaotong University, Zhaotong 657000, China

**Keywords:** biodiversity conservation, cultural significance, ethnobotanical surveys, *Gastrodia elata*, human well-being, sustainable development

## Abstract

This study aims to elucidate the intrinsic connections among Zhaotong’s *Gastrodia elata* (Tianma in Chinese) bioculture, the local ecological environment, and the development of the Tianma industry. Ethnobotanical surveys indicate that the Tianma industry, which has developed from the Tianma bioculture, has improved local livelihoods and boosted regional economic growth. The local government has actively engaged in forest conservation efforts and promoted reforestation projects, laying a solid ecological foundation for the sustainable utilization of Tianma. The sustainable development of the Tianma industry has alleviated poverty, preserved biodiversity, and driven economic growth, demonstrating how biocultural diversity enhances ecosystem services and human well-being.

## 1. Introduction

Biocultural diversity embodies the intricate interplay and synergistic relationship among biodiversity, cultural diversity, and linguistic diversity, forming the cornerstone for the sustainable development of both ecosystems and human societies [1,2,3]. Extensive research indicates that regions rich in biocultural diversity typically exhibit higher levels of biodiversity, richer cultural expressions, and enhanced human well-being [4,5,6]. Within this multidimensional framework, dietary and medicinal practices rooted in specific environments are particularly significant. Recent studies increasingly focus on plants serving dual purposes as food and medicine (medicinal–food homologous plants) and their associated biocultural practices, as these plants represent dynamic manifestations of biocultural diversity interactions [7,8,9].

As economies advance and living standards improve, heightened global health consciousness has shifted dietary preferences from prioritizing sufficiency and quality towards a demand for “green”, “pollution-free”, “natural”, and “medicinal–food homologous” products [10,11,12,13]. This trend has spurred the development of natural and healthy foods, particularly novel products utilizing medicinal–food homologous plants, which are increasingly favored by consumers [14,15]. China possesses a profound historical tradition in integrating medicinal and dietary uses, with ancient pharmacopoeias documenting numerous substances valued for both nourishment and therapy, highlighting their unique role in Traditional Chinese Medicine (TCM) health preservation and treatment [16,17]. *Gastrodia elata* Blume exemplifies this tradition.

*Gastrodia elata* Blume, known as Tianma in Chinese, an orchidaceous perennial parasitic herb, thrives in sparse forests, forest margins, and shrub clearings at altitudes of 400–3200 m [18,19]. Its dried tubers constitute a valuable TCM material, traditionally used to pacify liver wind, relieve convulsions, alleviate dizziness and headaches, and treat symptoms such as limb numbness, hemiplegia, and rheumatic pain [20,21]. Notably, the application of Tianma extends beyond medicinal purposes. Its culinary use has a long-standing history in Chinese folk culture. In November 2023, China’s National Health Commission (NHC), in conjunction with the State Administration for Market Regulation (SAMR), formally included Tianma in the Catalogue of Substances that are Both Food and Traditional Chinese Medicine under the Food Safety Law of the People’s Republic of China [22,23]. This regulatory recognition affirms its inherent “medicinal–food homologous” status and validates its longstanding public use.

Currently, commercial Tianma in China relies predominantly on cultivation, with major production areas forming the “Six Major Tianma Production Zones”: Zhaotong (Yunnan Province), Hanzhong (Shaanxi Province), Enshi (Hubei Province), various regions in Sichuan Province, Bijie and Kaili (Guizhou Province), and Jilin Province [24,25]. Among these, Zhaotong in Yunnan Province distinguishes itself through its extensive cultivation history, superior product quality, exceptional natural growing conditions, and advanced cultivation techniques, establishing preeminence domestically and international renown [26]. Its core status is further substantiated by official recognition: in 2004, “Zhaotong Tianma” was granted Protected Geographical Indication (PGI) status by the former General Administration of Quality Supervision, Inspection, and Quarantine (AQSIQ), and in 2014, Zhaotong was designated the “Hometown of Chinese *Gastrodia elata*” by the State Administration of Traditional Chinese Medicine (SATCM) [27,28].

Centuries of symbiotic interaction have fostered a rich traditional culture centered around Tianma among Zhaotong residents. Tianma functions not only as an indispensable medicinal resource but also as a distinctive culinary ingredient integral to local cuisine. As Zhaotong’s most iconic highland specialty, it vividly demonstrates how biocultural diversity enhances ecosystem services and promotes human well-being, serving as a model for harmonizing ecological conservation with socio-economic development. However, systematic scientific documentation and in-depth analysis of the rich biocultural diversity associated with Tianma in Zhaotong remains lacking. Therefore, this study employs ethnobotanical methodologies to conduct the first comprehensive survey in Yiliang County, the original production area of Zhaotong Tianma. The specific objectives are to (1) systematically document and analyze the traditional knowledge system surrounding Tianma in the Zhaotong region, (2) meticulously record ethnobotanical information pertaining to the plant, (3) quantitatively assess the cultural significance of species involved in Tianma cultivation and utilization, and (4) elucidate the dynamic interactions between Tianma-associated biocultural diversity, local ecological conservation, and economic development, thereby providing a scientific basis for sustainable utilization strategies.

## 2. Methods

### 2.1. Study Area

Zhaotong City, situated in northeastern Yunnan Province between 102°52′–105°19′ E and 26°55′–28°36′ N, is characterized by rugged terrain with deep valleys and steep mountains, typical of a mountainous region (Figure 1). The focal area of this study, Yiliang County, lies at the heart of Zhaotong City. Nestled within the Wumeng Mountains at the junction of Yunnan, Guizhou, and Sichuan provinces and upstream of the Yangtze River, it spans 279,576 hectares, bounded by 103°51′–104°45′ E and 27°15′–27°53′ N. The region experiences a highland monsoon climate with features of both subtropical and warm temperate zones. It is generally cool and humid all year round, with annual rainfall ranging from 960 to 1300 mm and relative humidity between 76% and 85%. These unique geographic and climatic conditions foster an ideal environment for diverse vegetation, including dry-hot sparse deciduous forests, wet-hot evergreen broad-leaved forests, semi-arid evergreen broad-leaved mixed forests, cool and humid evergreen broad-leaved and coniferous forests, and cold and humid evergreen broad-leaved forests [29]. The area boasts approximately 5000 species of higher plants, accounting for over one-third of the province’s total. It comprises 151 families, 457 genera, and more than 2000 woody plant species [30]. Among them, three genera and over twenty species are suitable as fungal materials for Tianma production. Moreover, the well-preserved primitive forests contribute to biodiversity conservation, and the natural humus soil offers an excellent growing environment for Tianma [31].

### 2.2. Ethnobotanical Investigation

Five ethnobotanical field surveys were conducted in Zhaotong City between August 2020 and May 2024. Eighteen survey sites (Figure 1) were selected across government departments, agricultural markets, Traditional Chinese Medicine clinics, research institutions, and traditional villages. To facilitate interviews, a structured questionnaire framework was developed a priori based on the 5W + 1H principle (Appendix A). The data collection employed free-listing, semi-structured interviews, and participatory observation [32,33].

Voucher specimens were collected during fieldwork. Vascular plant nomenclature followed Flora of China (www.iplant.cn/foc; accessed on 8–15 January 2024) and Plants of the World Online (https://powo.science.kew.org; accessed on 16–24 June 2025). Species identification was performed by Associate Professor Xiong Yong and Dr. Fan Yanxiao, with vouchers deposited in the Herbarium of the School of Ethnomedicine, Yunnan Minzu University. Survey data further indicate that *Armillaria* spp. and *Mycena* spp. fungal strains used by local farmers for Tianma cultivation were jointly supplied by the Zhaotong Municipal Government and Zhaotong Tianma Research Institute, with all strains technically authenticated by researchers.

The snowball sampling technique was adopted to identify 114 key informants. These informants comprised 63 males and 48 females, with an average age of 43 years [34]. To ensure the selection of highly relevant and knowledgeable participants, a rigorous screening standard was applied. Informants were required to have extensive familiarity with various facets of Tianma. They included Tianma cultivators, Chinese herbal medicine merchants, traditional herbalists, Tianma researchers, and government officials. In recognition of their valuable time and expertise, participants were offered modest compensation in the form of a small honorarium. During the interviews, the informants were queried about a wide range of aspects related to Tianma, both in historical and contemporary contexts. These included local names, uses, specific plant parts utilized, cultivation methods, harvesting times, preparation techniques, as well as sales and marketing practices.

Before initiating the investigation, we secured oral agreements from local administrative-territorial organizations (village administrations) and obtained oral free, prior, and informed consent (FPIC) from each participant. We consistently emphasized participants’ right to withdraw from the study at any time. This research strictly adheres to the ethical guidelines of the International Society of Ethnobiology (ISE).

### 2.3. Data Analysis

Data on the development of Zhaotong’s Tianma industry from 2017 to 2022 was compiled through three field visits to the Zhaotong Municipal Government. The dataset encompassed various parameters such as the planting area, fresh Tianma yield, planting output value, processing output value, and comprehensive output value. This data was utilized to gauge the influence of the Tianma industry on residents’ income. In addition, data on the changes in Zhaotong’s forest coverage was collected. This was conducted to assess the contribution of Tianma-related biocultural diversity to safeguarding the local ecological environment.

Moreover, the cultural importance index (*CI*) was computed to evaluate the cultural significance of the Species used in Tianma Cultivation (STC) [35]. The calculation formula is as follows:$$CI_{S}=\sum_{U=U_{1}}^{U_{NC}}\sum_{i=i_{1}}^{i_{N}}\frac{UR_{ui}}{N}$$

For a given STC denoted as *S*, let *N* be the total number of informants. *NC* represents the total number of usages for species *S*. *UR_ui_* stands for a utilization report (*UR*) of species *S* mentioned by the *i*th informant in usage *u*. The *CI* is derived by summing up the proportion of informants who mentioned each use purpose of a given species. This index reflects the extent to which a species is used and the diversity of its uses. Each mentioned use purpose holds significance for the plant’s importance. Consequently, a greater number of usages leads to a higher *CI* value.

## 3. Results

### 3.1. The Diversity of Species Used in Tianma Cultivation

The survey identified a total of 23 species (and forms) from seven families used in Tianma cultivation (Table 1), serving as either the target cultivated species, symbiotic fungi (promoting early-stage germination of Tianma), or fungus-cultivating wood. Fagaceae was the most dominant family, represented by 10 species. Local growers report that wild Tianma often grows in association with Fagaceae trees, as the symbiotic fungus *Armillaria* has evolved to efficiently utilize their wood. Fagaceae species (e.g., *Castanea mollissima* Blume, *Fagus longipetiolata* Seemen, and *Quercus glauca* Thunb.) have dense, starch-rich wood that decomposes slowly, providing *Armillaria* with a long-term, stable nutrient substrate. Crucially, Fagaceae trees are widely distributed in Zhaotong, allowing for easy local sourcing. The survey also noted that two Betulaceae species, *Betula alnoides* Buch.–Ham. ex D.Don and *Betula luminifera* H.J.P.Winkl. are used as substitutes when Fagaceae wood is scarce. However, key informants indicated that Betulaceae species contain less starch and provide weaker long-term nutrient support, potentially hindering Tianma’s later growth.

Three *Mycena* species (Mycenaceae), *Mycena dendrobii* F.C.Lin & C.R.Chien, *Mycena orchidicola* Fan & Guo, and *Mycena osmundicola* J.L.Maas are used to promote early stage Tianma germination. Among these, *M. osmundicola* is the most widely applied due to its stable performance in cultivation, achieving a germination rate of 70–90%, significantly higher than other fungi. *Armillaria mellea* (Vahl) P.Kumm. (Marasmiaceae) is essential for Tianma growth, as it decomposes wood to provide parasitic nutrition, facilitating tuber enlargement.

Apart from the original strain of *Gastrodia elata* (Tianma), local cultivators in the region further classify the cultivated Tianma into four distinct types based on tuber and inflorescence characteristics (shape, size, and color). Lü Tianma (绿天麻), Wu Tianma (乌天麻), Xuehong Tianma (血红天麻), and Huang Tianma (黄天麻) (Figure 2). Lü Tianma has long, oval, or obconical rhizomes, densely packed nodes, light blue–green stems, and light blue–green to white flowers. Wu Tianma has densely packed nodes on the rhizomes, gray–brown stems with white longitudinal stripes, and blue–green flowers. This type of Tianma has a high drying rate and excellent quality. Locals believe that Wu Tianma has relatively high levels of gastrodin, the main bioactive component among the four types of Tianma, making it the most commonly cultivated type in Zhaotong. The young shoots of Xuehong Tianma are red, the stem and flowers are light orange–red, the capsules are orange–red, and the tubers are stout and long-oval in shape. This variety grows quickly and has a high single-plant yield, making it widely cultivated in the region. Huang Tianma has long, strip-shaped tubers with dense nodes slightly twisted or curved, and stems and flowers that are lighter in color than Xuehong Tianma, appearing pale yellow.

Additionally, in Zhaotong’s traditional agroecological system, Tianma is often intercropped with corn and potatoes. Corn, being a tall crop, offers a shade that shields Tianma tubers from sunburn and dehydration. Potatoes, with their low-growing and spreading habit, suppress weeds by occupying ground space, and their shallow root systems reduce nutrient competition with Tianma tubers.

### 3.2. The CI Value of Species Used in Tianma Cultivation

As shown in Table 2, the calculated CI values for STC are as follows: *C. mollissima* (with a CI value of 2.29), *G. elata* (2.00), *G. elata* f. *flavida* (2.00), *G. elata* f. *glauca* (2.00), *G. elata* f. *viridis* (2.00), and Xuehong Tianma (2.00) exhibited relatively high cultural importance (CI) values. High CI values typically suggest that the informants are highly familiar with these plants and their various uses, indicating that these plants hold significant cultural importance within the community. For instance, *C. mollissima* is widely recognized by nearly all key informants in Zhaotong for its use as fungus-cultivating wood and as a food source. Similarly, the five species of Tianma are acknowledged by most key informants for their roles as both food and medicine, making them crucial cultural species with substantial cultural value in the local community.

On the other hand, *Quercus fabri Hance* (0.96), *Quercus aliena* Blume (0.94), *M. osmundicola* (0.90), *M. dendrobii* (0.72), *Quercus dentata* subsp. *yunnanensis* (Franch.) Menitsky. (0.70), *Castanea henryi* (Skan) Rehder & E.H.Wilson (0.68), *M. orchidicola* (0.68), *B. luminifera* (0.66), and *B. alnoides* (0.54) displayed low CI values. This likely implies that these plants have limited or no other uses apart from their role in Tianma cultivation or that few informants are aware of their other applications. However, these plants can still hold special significance for certain informants. For example, although *B. alnoides* may not be widely recognized for its uses, 38 informants mentioned its use as fungus-cultivating wood and 23 pointed out its use as timber, highlighting its value to those individuals.

### 3.3. Traditional Cultivating Methods of Tianma

Traditionally, Tianma can be cultivated using two primary methods: sexual reproduction through seed sowing and asexual reproduction by planting *Baitouma*, the primary tuber of Tianma, which has a distinct growth cone at the top. While it cannot reproduce sexually through stem pulling, it can form secondary tubers asexually.

For seed cultivation, Tianma seeds require symbiosis with two types of fungi, *Mycena* and *A. mellea*, to obtain nutrients. Before sowing, cultivated *Mycena* fungi are torn into pieces and placed in plastic containers. Tianma seeds are lightly sprinkled over the fungal pieces, ensuring even distribution to prevent concentration on a few pieces, which could hinder germination and inoculation. This process should be conducted indoors or in a sheltered area to prevent seed loss due to wind. Either pit planting or ridge planting is used in sandy soil conditions with shade cover. Holes are dug 20–25 cm deep and 70 cm wide (to accommodate fungal blocks), loosened at the bottom, and lined with 0.5 cm of damp tree leaves. The prepared fungal material is placed downhill in the pit, spaced approximately 4 cm apart, and covered with another thin layer of damp leaves before a final layer of 7–9 cm of sand. A top layer of pine needles ensures moisture retention, and the soil is thoroughly watered to maintain an initial humidity of 70% (Figure 3). After sowing and inoculation with symbiotic germination fungi, Tianma seeds germinate and form *Baitouma*, which can be transplanted in the winter or spring of the following year. These protocorms establish a nutritional relationship with *A. mellea*, initiating the first phase of asexual reproduction.

When using *Baitouma* for cultivation, the prepared fungal bed is uncovered, leaving the bottom layer of fungal wood intact while removing the upper fungal wood and branches. Tubers are sown at both ends of the fungal wood, with 1.5–2.0 kg of fungal branches placed, and a 3–5 cm layer of planting soil is applied. The tubers are placed close to the fungal wood, with about 50–60 g of tubers for each piece of fungal wood. This process is repeated to cultivate a second layer. In areas with abundant fungal wood, a third layer of new wood can be added for natural infection, to be used in the next planting season. After all planting is complete, a 10 cm layer of planting soil is added on top, followed by a layer of dead pine needles or straw for moisture retention.

### 3.4. The Management and Harvesting of Tianma

Field management of Tianma mainly encompasses moisture, temperature control, and pest and disease prevention. For moisture management, it is crucial to prevent both drought and waterlogging. This involves regularly monitoring soil humidity, maintaining air humidity within the range of 70–90% and soil humidity at 40–50%. According to the informants, humidity control during the autumn is vital to curb the excessive growth of *A. mellea*, which could deplete the nutrients in the wood and negatively impact Tianma. During the rainy season, steps should be taken to avoid waterlogging, while in dry periods, the soil should be kept moist. Temperature management focuses on preventing both high temperatures and frost, with regular checks on soil temperature. Tianma tubers germinate at 15 °C and grow optimally at around 25 °C. In contrast, *A. mellea* thrives between 6 and 8 °C, grows fastest at 20–25 °C, and ceases growth above 30 °C. Shading should be employed to mitigate high temperatures. Pest and disease control should prioritize prevention, applying appropriate treatments as required.

In Zhaotong, Tianma is typically harvested and planted at the same time. The harvesting season spans two periods: winter and spring. Winter harvesting occurs before the ground freezes, usually from November to February the following year. Harvesting too early leads to high water content in the tubers, resulting in poor quality after drying. Harvesting too late exposes the tubers to cold air, risking frost damage. Spring harvesting takes place after the ground thaws, typically from March to May. It should be carried out after the thaw but before the tuber buds sprout. Delayed harvesting can cause the emergence of flower stalks, which consume significant nutrients, reducing the weight and quality of the tubers. According to key informants, Tianma harvested in the winter is of the highest quality, with the greatest content of active ingredients. However, in the past, Tianma for medicinal use was predominantly harvested in the spring because herbalists could only discover it after it sprouted in the spring, leading to inferior quality compared to winter-harvested Tianma.

Under natural conditions, sunny days are optimal for harvesting, as they facilitate operations, prevent mud from adhering to the tubers, and ensure high quality and storability. Manual digging is the common method, starting with the removal of topsoil. Care must be taken to avoid damaging the tuber buds. When approaching the fungal wood, a hoe is used to carefully pry it out and extract the tubers. Tubers tightly wedged between two pieces of fungal wood require careful handling to prevent breakage. After removing the fungal wood, the surrounding soil should be inspected for any missed tubers, particularly on the uphill side.

### 3.5. Traditional Pharmacological Effects and Processing of Tianma

In Zhaotong, Tianma is commonly prepared in two forms. It can be powdered for individual consumption or combined with herbs such as the dried root of *Hansenia weberbaueriana* (Fedde ex H. Wolff) Pimenov & Kljuykov, *Heracleum hemsleyanum* Diels, *Angelica sinensis* (Oliv.) Diels, and *Rehmannia glutinosa* (Gaertn.) Libosch. ex Fisch. & C. A. Mey. and then mixed with honey to form medicinal pills. Taking Tianma powder alone offers notable benefits for cardiovascular diseases. On the other hand, Tianma pills are effective in dispelling wind and dampness, clearing meridians, alleviating pain, and nourishing the liver and kidneys, thereby treating conditions like limb spasms, numbness in hands and feet, and lower back and leg pain.

Regarding the processing of Tianma, residents in Zhaotong possess unique insights. Traditional preliminary processing methods mainly involve steaming Tianma in a pot (Figure 4), followed by direct open-fire drying. During the drying process, the herb must undergo “sweating” three to four times. The “sweating” method, a common traditional Chinese medicinal processing technique, involves heating or semi-drying fresh herbs, then piling them up in a sealed manner to generate heat, causing internal moisture to evaporate and condense into droplets on the surface, resembling human sweating. This traditional method is time-consuming and labor-intensive, and sulfur fumigation is often employed to prevent mold during drying, which can adversely affect the medicinal efficacy of Tianma. Locals believe this method ensures long-term preservation without spoilage. With the development of the Tianma industry, traditional processing methods have gradually been replaced by more efficient techniques such as sun-drying, shade-drying, oven-drying, and vacuum freeze-drying. However, it remains uncertain which method best preserves the quality and efficacy of Tianma. Ensuring the sustainable development of the Tianma medical industry in Zhaotong, standardizing processing techniques, and guaranteeing product quality are urgent tasks.

### 3.6. Culinary Culture and Modern Integration of Tianma

Tianma is prepared in various ways to cater to different tastes in Zhaotong. Common cooking methods include stewing with chicken or pork, adding it to hot pots, stir-frying, and steaming. Local specialties like “Steamed Chicken with Tianma”, “Tianma and Black Bean Tofu Soup”, “Sweet and Sour Stir-Fried Filamentous Tianma”, and “Cold Tianma Sprouts Salad” (Figure 5, Table 3) showcase both the herb’s flavor and health benefits. For example, to make “Steamed Chicken with Tianma”, a black-boned chicken is slaughtered, cleaned, chopped, blanched, and then steamed with Tianma, scallions, ginger, Sichuan peppercorns, cooking wine, and salt. This dish is not only nutritious but also invigorates the body and promotes liver and eye health, making it suitable for those recovering from illness or experiencing dizziness, blurred vision, or limb numbness. It is a favorite among both locals and tourists.

Cooking with Tianma requires skill, aesthetic sense, and a harmonious blend of color, aroma, taste, shape, and presentation, offering a delightful sensory experience. The naming and tasting of Tianma dishes in Zhaotong are carefully crafted, reflecting ingredients, cooking methods, or historical, mythical, and anecdotal references. The culinary culture of Zhaotong Tianma is closely tied to local lifestyles and customs, often celebrated through festivals and folk traditions. For instance, the Tianma Harvest Festival, held in the autumn and winter, highlights the harvesting and processing of Tianma, featuring culinary competitions centered around the herb, attracting many visitors.

With the development of modern dietary culture, Tianma has been innovatively incorporated into various food products, such as Tianma tea, beverages, and dietary supplements. These products retain Tianma’s medicinal benefits while meeting the demands of a fast-paced modern lifestyle, offering convenience and portability. The culinary culture of Tianma not only enriches the dining tables of Zhaotong residents but also helps build the local Tianma brand, fosters the Tianma gourmet economy, attracts tourists and food enthusiasts, and promotes local economic development.

### 3.7. Ecological Protection and Sustainable Use of Tianma

As the Tianma industry has grown rapidly, the demand for wood used in Tianma cultivation has surged, leading to an increasing scarcity of this vital resource. To protect the delicate ecological environment necessary for Tianma’s growth and ensure a sustainable supply of high-quality wood, the local government has implemented a series of protective measures, focusing on safeguarding woodlands and promoting ecological balance.

A key strategy is to integrate Tianma woodland development into broader forestry projects. The municipal forestry and grassland departments have issued policies allowing enterprises, cooperatives, and large-scale farmers involved in Tianma woodland construction to apply for afforestation projects. These projects are given priority in funding allocations, including subsidies for returning farmland to forests and forest vegetation restoration fees.

Since 2021, the Zhaotong Municipal Government has actively responded to the need for forest quality improvement by launching multiple ecological projects. Some of these specifically target Tianma woodlands, such as forest tending (59,133.33 hectares), degraded forest restoration (3633.33 hectares), and low-efficiency forest transformation (45,333.33 hectares). These efforts have achieved significant results: Zhaotong’s forest coverage rate increased from 32.98% in 2017 to 50.10% in 2022. This remarkable growth highlights the Tianma industry’s role not only as an economic driver but also as a contributor to ecological conservation. Surveys conducted reveal that, by 2023, Tianma cultivation had expanded across seven counties (districts), 41 towns, and 92 administrative villages in Zhaotong City, benefiting 22,300 households and involving a total of 100,200 individuals. Notably, the thriving Tianma industry has played a pivotal role in poverty alleviation, lifting 6100 households and 25,600 individuals out of poverty, while contributing to an average per capita income increase of CNY 5678.

### 3.8. Development and Market Expansion of Tianma Industry

The Zhaotong Municipal Government has vigorously promoted the development of the Tianma industry through policy support and technological promotion, leading to a continuous expansion of the planting area and production volume of Tianma in Zhaotong. Currently, Zhaotong is one of the six major Tianma production areas in China.

The government promotes a “company + cooperative + base + farmer” development model, advancing the construction of high-standard demonstration planting bases. This model emphasizes the provision of high-quality seeds, “two fungi” (*Mycena* genus and *A. mellea*), and fungal wood, and widely promotes under-forest semi-wild cultivation techniques. The aim is to enhance the scientific, large-scale, and standardized level of Tianma planting, synchronizing the expansion of the planting scale with improvements in quality and efficiency.

In terms of product distribution, the Zhaotong Municipal Government has improved infrastructure and established the first professional comprehensive market for Tianma trading in China, the Yiliang Xiaocaoba International Tianma Trading Center. The government also attracts well-known pharmaceutical companies to invest in the Tianma industry, vigorously supporting the growth of enterprises, cooperatives, and individual businesses, forming an integrated industry system covering planting, processing, and sales.

Currently, Zhaotong has 863 business entities in the Tianma industry, including 239 enterprises, 567 farmers’ cooperatives, and 57 other business entities. The city has cultivated one national key leading enterprise, two provincial-level leading enterprises, and ten municipal-level leading enterprises. The products fall into four main categories, medicines, health products, food, and daily chemicals, with 32 different products and 19 brands (Figure 6). These products are distributed in 143 prefecture-level cities across 32 provinces (autonomous regions and municipalities) in China, and Zhaotong Tianma toothpaste products are exported to South Korea, Vietnam, Thailand, Dubai, and other regions.

## 4. Discussion

Folk taxonomy serves as a traditional knowledge system utilized by specific cultural or ethnic groups for biological identification, classification, nomenclature, and related activities [36,37]. This locally derived classification framework integrates biological attributes with cultural elements [38,39]. Substantial correspondence exists between folk and scientific classification systems, particularly in biodiverse regions with high linguistic diversity, where folk classification contributes significantly to biodiversity documentation and traditional knowledge preservation [40,41]. This study confirms a notable consistency between the folk classification and scientific classification of Zhaotong Tianma. Field investigations and cultivation records confirm that locally identified Lü Tianma, Wu Tianma, and Huang Tianma correspond to *G. elata* f. *viridis* (Makino) Makino, *G. elata* f. *glauca* S. Chow, and *G. elata* f. *flavida* S. Chow, respectively, while the Xuehong Tianma may belong to *G. elata* f. *elata*, requiring further molecular verification. Taxonomically, all these forms (f.) are heterotypic synonyms of *G. elata*. [42,43]. Genomic analyses indicate differential intraspecific genetic diversity, with *G. elata* f. *elata* exhibiting the highest diversity and *G. elata* f. *viridis* exhibiting the lowest [44]. Despite widespread application in cultivation and commerce, precise varietal identification faces challenges due to hybridization, regional variations, and post-harvest morphological alterations.

Pharmacological research provides scientific support for traditional medicinal applications. Tianma extracts demonstrate four principal bioactivities: central nervous system sedation/anticonvulsion/analgesia, cardiovascular cardiotonic/hypotensive effects, anti-hypoxic capacity, and immune modulation [22,44,45]. The primary constituent gastrodin mitigates myocardial ischemia–reperfusion injury through increased coronary flow, the suppression of endothelin-1 and pro-inflammatory cytokines (TNF-α/IL-6), oxidative stress reduction, and apoptosis inhibition [46,47]. Multi-omics analyses further elucidate mechanisms: metabolomics identifies 15 blood–brain barrier permeable components (notably gastrodin and parishins A/C) activating hippocampal neurogenesis via EGFR-PI3K/AKT signaling [48]; probiotic-fermented secondary liquid (SFL) enhances gastrodin bioavailability while modulating dopamine receptor d1 and DNA repair genes (msh2, recql4) to improve sleep patterns [49]; and gastrodia polysaccharide (GEP) coordinates oxidative stress, heat shock response, and endoplasmic reticulum stress pathways against Alzheimer’s Disease (AD) [50]. While Phase III clinical trials remain pending, preclinical evidence supports therapeutic potential in cardiovascular and neurological protection, particularly through gut–brain axis regulation, antioxidative, and anti-inflammatory pathways, substantiating traditional applications for cerebro–cardiovascular conditions.

The “medicinal and edible homology” characteristic of Zhaotong Tianma further enhances its socio-economic value [51]. Research confirms that Zhaotong Tianma contains higher gastrodin content than that from other regions. This significant advantage grants it greater market competitiveness and constitutes the primary reason for its generally higher price compared to Tianma from other areas [52]. Long-term consumption of Zhaotong Tianma is considered beneficial for alleviating fatigue, enhancing immunity, and improving memory [53,54]. Precisely this unique nutritional value and multifunctionality have led to Zhaotong Tianma being highly valued in the local dietary culture. However, the active ingredient content varies among different variants of Tianma within the Zhaotong region. For example, studies show that Huang Tianma has the highest gastrodin content (14,500 μg/g), followed by Wu Tianma (6000 μg/g), with Lü Tianma having the lowest (5400 μg/g). This finding appears inconsistent with the widely held view in the Zhaotong region that Wu Tianma possesses the best quality, warranting further research [55,56]. Nowadays, the flourishing Zhaotong Tianma industry is attracting a large number of young people, who recognize its livelihood value, to return to their hometowns for employment. These young people actively participate in all segments of the industrial chain, including cultivation, medicinal trade, scientific research, product processing, and resource conservation. They thus secure stable local incomes without needing to seek work far from home. This trend of intergenerational shift is strongly corroborated by the demographic characteristics of the 114 key informants (average age: 43) surveyed in this research.

Despite the meaningful work conducted in this study, numerous relevant and interesting research avenues remain to be explored. For instance, the cultivation of Tianma heavily relies on the symbiotic *A. mellea*, yet current research has not delved deeply into how to manage *A. mellea* cultivation to prevent pathogen spread into natural ecosystems or disrupt forest ecological balance (e.g., affecting local soil microbiomes). Secondly, although Zhaotong has implemented reforestation plans, research has not yet assessed the long-term potential impacts of wood-intensive cultivation modes (requiring substantial amounts of wood for *A. mellea* growth) on forest ecosystem integrity, biodiversity, and carbon sequestration capacity. Additionally, the transformations that commercialization may bring to traditional knowledge transmission modes, as well as the true net ecological footprint of the entire industry, also require more comprehensive longitudinal assessments.

In summary, the case of Zhaotong Tianma exemplifies a distinctive bioresource management practice that integrates traditional wisdom and drives socio-economic vitality. This practice offers a potentially valuable demonstration model for other regions in Asia and beyond seeking to leverage traditional knowledge, biodiversity, and sustainable agriculture to promote community development and cultural heritage. The key to the success of this model lies in the synergistic realization of deeply linking biocultural diversity conservation with the Sustainable Development Goals (SDGs) (Table 4). However, whether this model can truly succeed and be widely replicable hinges significantly on effectively addressing the identified ecological limitations, particularly the safe management of *A. mellea* and the long-term ecological impacts of wood-resource-intensive cultivation modes, to ensure the industry’s development towards ecological sustainability.

## 5. Conclusions

This study presents a successful model demonstrating that the biocultural diversity associated with high cultural significance species, such as *Gastrodia elata* in Zhaotong, holds substantial value for advancing human well-being, ecological conservation, and economic development. Ethnobotanical investigations revealed a complex network of species supporting Tianma cultivation, documenting 23 species across seven botanical families utilized for symbiotic fungi cultivation, fungus-hosting timber, and cultivation substrates, with Fagaceae representing the dominant family. Cultural importance (CI) analysis identified *Castanea mollissima* and various Tianma morphotypes as culturally significant taxa, while other species exhibited lower CI values indicative of specialized or limited applications. Traditional propagation techniques were documented, encompassing both sexual reproduction via seed sowing and asexual propagation using “*Baitouma*” (the primary tuber of Tianma), emphasizing the essential role of symbiotic fungi including *Mycena* spp. and *Armillaria mellea*. Field management protocols involving precise moisture regulation, temperature control, and pest prevention proved critical for optimizing growth. Harvesting protocols required meticulous winter or spring collection to preserve medicinal quality. The research further elucidated traditional pharmacological applications, processing methodologies, culinary traditions, and integration into contemporary functional foods, highlighting the species’ versatility and commercial potential. Ecological sustainability measures included incorporating Tianma woodland management into broader forestry initiatives to ensure a sustainable timber supply. Policy interventions by the Zhaotong Municipal Government including regulatory support, technological dissemination, and infrastructure investment have driven significant expansion in the cultivation area and production volume, establishing Zhaotong as China’s primary Tianma production region. The industry now encompasses diverse product lines distributed domestically and exported internationally.

Looking ahead, the Zhaotong Municipal Government should take measures to attract more outstanding scientific and technological talents for the deep processing of Tianma. It should also enhance the recognition of the pivotal role of the Tianma processing industry in the overall development of the Tianma sector, focusing on the holistic layout of the entire industrial chain. By stimulating both demand and technological progress, the government can promote the efficient and comprehensive utilization of resources, fostering all-around development.

## Figures and Tables

**Figure 1 biology-14-00846-f001:**
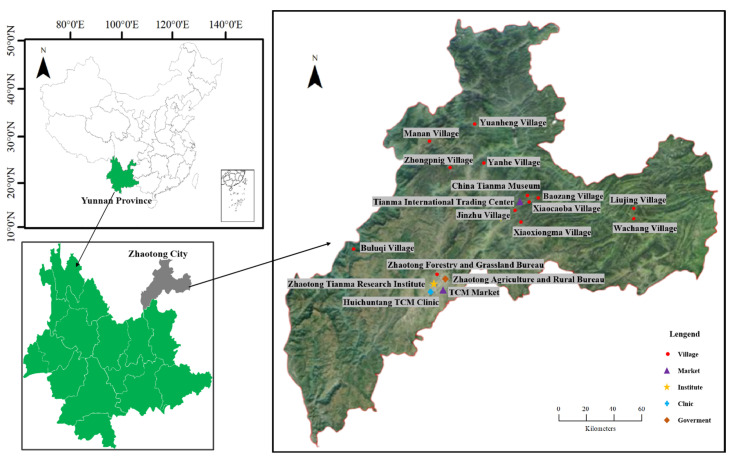
Research sites in Zhaotong City, Northeastern Yunnan Province, China.

**Figure 3 biology-14-00846-f003:**
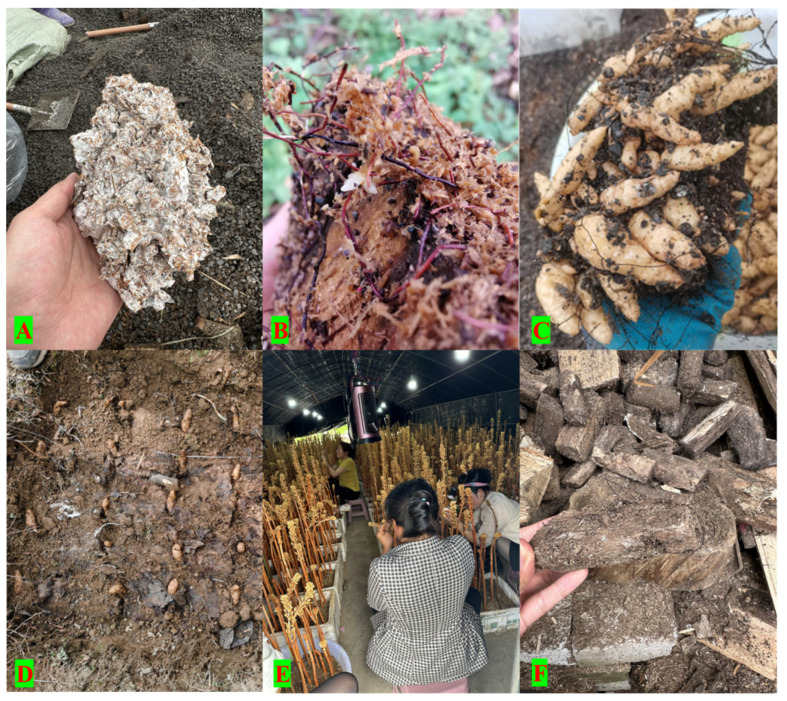
The planting of Tianma in Xiaocaoba Town, Yiliang County, Zhaotong City ((**A**): germination fungi (*Mycena* genus) for Tianma seeds; (**B**): *Armillaria mellea* (Vahl) P. Kumm; (**C**): *Baitouma* formed from Tianma seeds; (**D**): asexual reproduction through the planting of *Baitouma*; (**E**): Tianma’s pollination; (**F**): the utilized fungal wood can also be used as firewood).

**Figure 4 biology-14-00846-f004:**
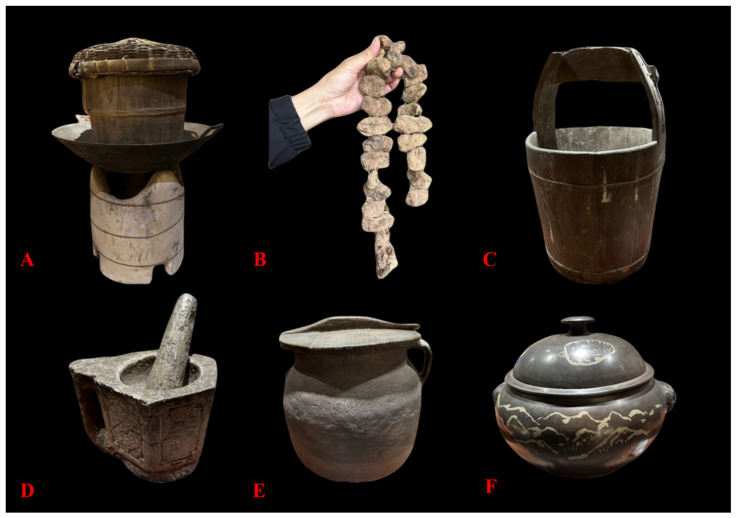
Tools traditionally used by indigenous people in Zhaotong for the processing of Tianma ((**A**): steamer and earthen stove; (**B**): Tianma processed by traditional methods; (**C**): wooden barrel used to store fresh Tianma; (**D**): stone mortar used for pounding Tianma powder; (**E**): pottery jar used to store processed Tianma; (**F**): clay pot used to boil Tianma).

**Figure 5 biology-14-00846-f005:**
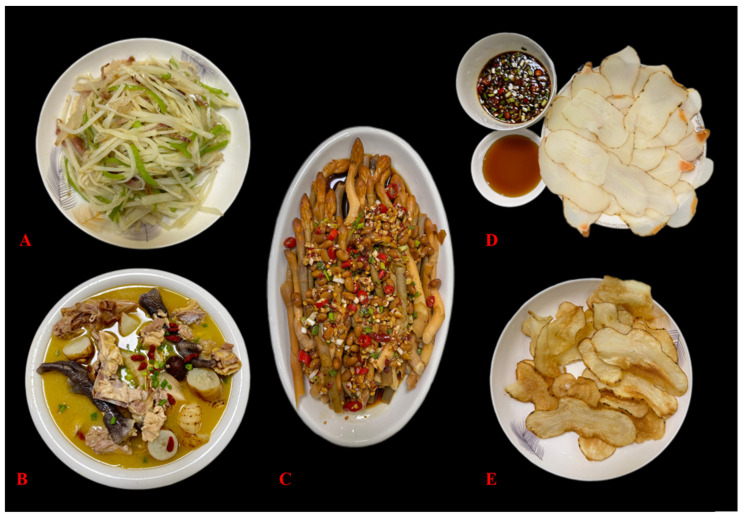
Local specialty dishes made with Tianma ((**A**): sweet and sour stir-fried filamentous Tianma; (**B**): steamed chicken with Tianma; (**C**): cold Tianma sprouts salad; (**D**): chilled fresh Tianma; (**E**): fried Tianma chips).

**Figure 6 biology-14-00846-f006:**
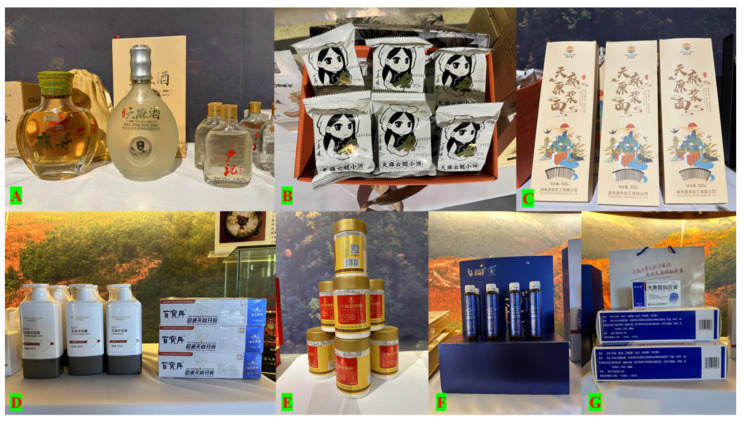
Various Tianma products ((**A**): Tianma baijiu; (**B**): Tianma snacks; (**C**): Tianma noodles; (**D**): Tianma daily chemical products; (**E**–**G**): Tianma medicine ((**E**): powder; (**F**): drinks; (**G**): capsules)).

**Figure 2 biology-14-00846-f002:**
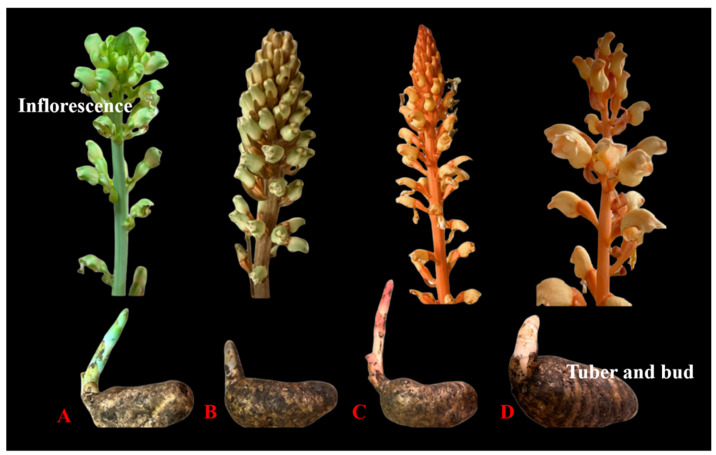
Four forms of Tianma cultivated in Zhaotong ((**A**): Lü Tianma; (**B**): Wu Tianma; (**C**): Xuehong Tianma, and (**D**): Huang Tianma).

**Table 1 biology-14-00846-t001:** Species diversity in the cultivation of Tianma in Zhaotong.

Family	Species	Chinese Name	Purpose	Characteristics	Specimen No.
Betulaceae	*Betula alnoides* Buch.–Ham. ex D.Don	西南桦 Xinanhua	Fungus-cultivating wood	Straight-grained wood; good *Armillaria* adaptability.	ZT013
	*Betula luminifera* H.J.P.Winkl.	亮叶桦 Liangyehua	Fungus-cultivating wood	Glossy leaves; lightweight wood with strong *Armillaria* affinity.	ZT010
Fagaceae	*Castanea henryi* (Skan) Rehder & E.H.Wilson	锥栗 Zhuili	Fungus-cultivating wood	Smooth bark; rapid *Armillaria* colonization.	ZT015
	*Castanea mollissima* Blume	板栗 Banli	Fungus-cultivating wood	Gray–brown fissured bark; hard and decay-resistant; supports *Armillaria* growth.	YL002
	*Castanea seguinii* Dode	茅栗 Maoli	Fungus-cultivating wood	Dark gray bark; high-density wood with extreme decay resistance.	YL001
	*Fagus sinensis Oliv.*	水青冈 Shuiqinggang	Fungus-cultivating wood	Alternative wood source in some regions.	ZT012
	*Quercus acutissima* Carruth.	麻栎 Mali	Fungus-cultivating wood	Deeply fissured bark; sharply serrated leaf margins; dense and decay-resistant wood.	ZT006
	*Quercus aliena* Blume	槲栎 Huili	Fungus-cultivating wood	Large wavy-margined leaves; hard and decay-resistant; suitable for high humidity.	ZT008
	*Quercus fabri* Hance	白栎 Baili	Fungus-cultivating wood	Gray–white pubescent leaves; fine-grained wood; rapid *Armillaria* colonization.	ZT009
	*Quercus glauca Thunb.*	青冈 Qinggang	Fungus-cultivating wood	Evergreen species: dense wood, stable *Armillaria* growth, and suitable for long-term cultivation.	ZT018
	*Quercus variabilis* Blume	栓皮栎 Shuanpili	Fungus-cultivating wood	Thick cork-like bark; drought-resistant; supports *Armillaria* growth.	ZT007
	*Quercus dentata subsp. yunnanensis (Franch.) Menitsky*	云南波罗栎 Yunnanboluoli	Fungus-cultivating wood	Endemic to Yunnan; cold- and barren-tolerant; used in high-altitude plantations.	ZT011
Marasmiaceae	*Armillaria mellea* (Vahl) P.Kumm.	蜜环菌 Mihuanjun	Symbiotic fungus	Essential for *Gastrodia* germination and growth (parasitic nutrition).	YL007
Mycenaceae	*Mycena dendrobii* F.C.Lin & C.R.Chien	石斛小菇 Shihuxiaogu	Symbiotic fungus	Promotes early-stage *Gastrodia* germination; less common than *M. osmundicola*.	YL005
	*Mycena orchidicola* Fan & Guo	兰小菇 Lanxiaogu	Symbiotic fungus	Enhances germination and seedling vigor; used experimentally.	YL006
	*Mycena osmundicola* J.L.Maas	紫萁小菇 Ziqixiaogu	Symbiotic fungus	Synergistic with *Armillaria*; high germination rates in Zhaotong.	YL004
Orchidaceae	*Gastrodia elata* Blume	天麻 Tianma	Target cultivated species	Core crop; tubers contain gastrodin and other bioactive compounds.	ZT001
	*Gastrodia elata* f. *flavida* S.Chow	黄天麻 Huang Tianma	Target cultivated species (variant cultivar)	Yellow tubers; drought-resistant; lower-altitude adaptation.	ZT003
	*Gastrodia elata* f. *glauca* S.Chow	乌天麻 Wu Tianma	Target cultivated species (variant cultivar)	Jade-like tubers; cold-tolerant; disease-resistant but slow-growing (2–3 years).	ZT005
	*Gastrodia elata* f. *viridis* (Makino) Makino	绿天麻 Lü Tianma	Target cultivated species (variant cultivar)	Greenish tubers (chlorophyll); short cycle (1.5–2 years) but pest-prone.	ZT004
	*Gastrodia* sp.	血红天麻 Xuehong Tianma	Target cultivated species (variant cultivar?)	Reddish tubers; lower medicinal value; prone to hollowing.	ZT002
Poaceae	*Zea mays* L.	玉米 Yumi	Intercrop/rotation crop	Provides shade and moisture retention; reduces soil degradation.	YL003
Solanaceae	*Solanum tuberosum* L.	马铃薯 Malingshu	Intercrop/rotation crop	Improves microclimate; mitigates continuous cropping issues.	YL008

Species in the inventory are arranged alphabetically by the family name. All these forms (f.) are heterotypic synonyms of *Gastrodia elata* Blume.

**Table 2 biology-14-00846-t002:** Cultural importance index of Species used in Tianma Cultivation.

Species	Purpose					CI
	Fungus-Cultivating Wood	Food	Material	Firewood	Others	
*Castanea mollissima* Blume	102	114	11	32	2 (ME)	2.29
*Gastrodia elata* Blume		114	114			2.00
*Gastrodia elata* f. *flavida* S.Chow		114	114			2.00
*Gastrodia elata* f. *glauca* S.Chow		114	114			2.00
*Gastrodia elata* f. *viridis* (Makino) Makino		114	114			2.00
Xuehong Tianma (*Gastrodia* sp.)		114	114			2.00
*Quercus acutissima* Carruth.	111		28	67		1.81
*Solanum tuberosum* L.		114			89 (IC)	1.78
*Zea mays* L.		114			85 (IC)	1.75
*Fagus sinensis* Oliv.	92		20	57		1.48
*Quercus variabilis* Blume	79		24	53		1.37
*Quercus glauca* Thunb.	63		55	28		1.28
*Castanea seguinii* Dode	88	21		33		1.25
*Armillaria mellea* (Vahl) P.Kumm.					114 (SF)	1.00
*Quercus fabri* Hance	68		14	27		0.96
*Quercus aliena* Blume	73			34		0.94
*Mycena osmundicola* J.L.Maas					103 (SF)	0.90
*Mycena dendrobii* F.C.Lin & C.R.Chien					82 (SF)	0.72
*Quercus dentata* subsp. *yunnanensis* (Franch.) Menitsky	52		8	20		0.70
*Castanea henryi* (Skan) Rehder & E.H.Wilson	64			13		0.68
*Mycena orchidicola* Fan & Guo					77 (SF)	0.68
*Betula luminifera* H.J.P.Winkl.	41		15	19		0.66
*Betula alnoides* Buch.–Ham. ex D.Don	38			23		0.54

ME: medicine; SF: symbiotic fungus; IC: intercrop/rotation crop; CI: cultural importance index. Ranked by CI values from high to low. All these forms (f.) are heterotypic synonyms of *Gastrodia elata* Blume.

**Table 3 biology-14-00846-t003:** Local specialty dishes that use Tianma as the main ingredient (partial list).

Dish Name	Part Used	Cooking Method	Seasonings (Ingredients)	Seasonings (Products)	Accompaniments
Steamed chicken with Tianma	Tuber	Steamed	Green onion, ginger, Sichuan pepper	Salt, cooking wine, etc.	Silkie chicken meat
Tianma and black bean tofu soup	Tuber	Boiled	Red pepper, Sichuan pepper	Salt, MSG, soybean oil	Black bean curd, green pepper
Sweet and sour stir-fried filamentous Tianma	Tuber	Stir-fried	Green onion, red pepper, garlic	Salt, MSG, soybean oil, aged vinegar, etc.	Green pepper
Stir-fried Tianma sprouts	Sprouts	Stir-fried	Green onion, ginger, garlic	Salt, MSG, soybean oil, etc.	/
Cold Tianma sprouts salad	Sprouts	Blanched, cold dressed	Green onion, garlic, ginger, small chili peppers	Sesame oil, soy sauce, aged vinegar, salt, MSG, etc.	/
Cold Tianma shreds	Tuber	Cold dressed	Green onion, garlic, ginger, small chili peppers	Sesame oil, soy sauce, aged vinegar, salt, MSG, etc.	/
Chilled fresh Tianma	Tuber	Sliced, blanched, chilled	/	Honey, spicy chili powder	/
Fried Tianma chips	Tuber	Deep-fried	/	Soybean oil, pepper, salt/honey	/
Steamed eggs with Tianma	Tuber	Steamed	/	Rock sugar	Eggs
Tianma and crucian carp soup	Tuber	Stewed	Ginger	Cooking wine, salt, MSG, pepper powder, vegetable oil, etc.	Crucian carp, red dates, goji berries
Tianma honey Cream	Tuber	Boiled	/	Honey, Tianma powder	/

/: no need.

**Table 4 biology-14-00846-t004:** Linkage analysis between zhaotong tianma and sustainable development goals.

Dimension	Associated SDGs	Core Synergistic Value
Cultural Heritage and Knowledge Preservation	SDG 4.7, 11.4	Scientifically documenting and transmitting traditional Tianma knowledge enhances cultural identity, promotes integrating local knowledge into education systems (SDG 4.7), and safeguards intangible cultural heritage (SDG 11.4).
Sustainable Utilization of Biodiversity	SDG 2.5, 12.2, 15	Ensuring safe management of *Armillaria* fungi and the sustainable supply of forest resources maintains agricultural biodiversity (SDG 2.5), achieves responsible resource management (SDG 12.2), and protects terrestrial ecosystems (SDG 15).
Inclusive Economic Growth and Employment	SDG 8.1, 8.5, 8.8	The industry chain creates diversified local livelihoods (SDG 8.5), reduces out-migration for work (SDG 8.8), and fosters inclusive regional economies (SDG 8.1).
Health Promotion	SDG 3	Scientifically validated medicinal/dietary values directly contribute to disease prevention and enhanced health and well-being (SDG 3).
Green Production and Consumption	SDG 12.2, 12.8	Promoting ecologically friendly cultivation and processing guides market choices towards sustainably certified products (SDG 12.8) and reduces the industry’s environmental footprint (SDG 12.2).

## Data Availability

The data presented in this study are available upon request from the corresponding authors.

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
