# Peer review of "Exploring the Biocultural Nexus of *Gastrodia elata* in Zhaotong: A Pathway to Ecological Conservation and Economic Growth"

_biology, 2025, doi:10.3390/biology14070846_

Round 1

Reviewer 1 Report (Previous Reviewer 1)

Comments and Suggestions for Authors
  • L15-18: A very long introduction about the plant being studied, its place is in the introduction section, preferably an introductory sentence that includes the novelty of the current study.
  • Choose expressive and important keywords, while not repeating any word mentioned in the title of the manuscript. Plz write the keywords in alphabetical order. Also, plz capitalize the first letter of each word.
  • L73-77: plz try to abbreviate this part
  • Introduction: flow is lost in the middle. Please connect all portions to make a continuous story and reader friendly.
  • Introduction: The hypothesis part is weak, please improve it.
  • Results and discussion should be separated. They are not givinga  clear picture in the joint form.
  • L484: Avoid using the personal pronouns in the entire manuscript.
  • Conclusion: Conclusions; do not repeat the above sentences in the conclusion part. In conclusion, you should write a summary of your work in short sentences so that I, as a reader of the reference article, can understand what the article ended up being. Add future perspectives at the end of conclusion.
  • Please use third person language throughout the manuscript.

Comments on the Quality of English Language

The English could be improved to more clearly express the research.

Author Response

Responses to Reviewer 1:

Comments 1: L15-18: A very long introduction about the plant being studied, its place is in the introduction section, preferably an introductory sentence that includes the novelty of the current study. Choose expressive and important keywords, while not repeating any word mentioned in the title of the manuscript. Plz write the keywords in alphabetical order. Also, plz capitalize the first letter of each word.

Response 1: As suggested, the overly lengthy introductory description was removed. The section now prioritizes an opening sentence highlighting the novelty of the current study. Expressive and important keywords were selected, avoiding repetition of words in the manuscript title, listed in alphabetical order, and with the first letter of each word capitalized. Please see Lines 15-23 and 49-50 for details.

Comments 2: L73-77: plz try to abbreviate this part

Response 2: Following your recommendation, this section has been abbreviated. Descriptions of Gastrodia elata's morphology and some redundant expressions were removed. Please see Lines 73-78 for the revised details .

Comments 3: Introduction: flow is lost in the middle. Please connect all portions to make a continuous story and reader friendly.Introduction: The hypothesis part is weak, please improve it.

Response 3: In accordance with your suggestions, the Introduction has been rewritten without altering the original meaning, focusing on:

(1) Providing a clearer explanation of the biocultural diversity concept and its link to sustainable development, naturally leading to "medicine-food homology plants" as the research focus to introduce the Gastrodia case study.

(2) Creating a tighter logical connection: after describing shifting consumption trends, directly highlighting China's long history of medicine-food homology to introduce Gastrodia as a representative example.

(3) Integrating Gastrodia's botanical characteristics, traditional medicinal value, its "edible" attribute, and recent regulatory recognition into a cohesive unit.

(4) Strengthening the presentation of the hypothesis.

(5) Maintaining an objective tone by removing any potentially emotional language from the original Chinese draft.

Please see Lines 52-115 for the revised Introduction.

Comments 4: Results and discussion should be separated. They are not givinga clear picture in the joint form.

Response 4: As advised, the Results and Discussion sections have been separated to allow for a clearer presentation of the study's core findings. Results are presented in Lines 192-430, and the Discussion in Lines 431-512.

Comments 5: L484: Avoid using the personal pronouns in the entire manuscript. 

Conclusion: Conclusions; do not repeat the above sentences in the conclusion part. In conclusion, you should write a summary of your work in short sentences so that I, as a reader of the reference article, can understand what the article ended up being. Add future perspectives at the end of conclusion. 

Please use third person language throughout the manuscript.

Response 5: Following your guidance:

  • The Conclusion section has been rewritten. The revision focuses on concisely summarizing the study's core findings in short sentences, enabling readers to quickly grasp the article's value.
  • Future research perspectives have been added at the end of the Conclusion. Please see Lines 513-545for details.
  • The entire manuscript has been thoroughly checked to ensure consistent use of the third person, eliminating personal pronouns.

Thank you very much for your insightful comments and suggestions, which have significantly improved the quality and clarity of our manuscript.

Reviewer 2 Report (Previous Reviewer 2)

Comments and Suggestions for Authors
    • Have molecular techniques been applied to confirm the identity of Gastrodia varieties or Mycena and Armillaria species? 
    • Could folk classifications be supported or contrasted by genetic data? 
    • Were gastrodin and other bioactives quantified across the different forms of G. elata? 
    • Is there a local perception of which variety is most medicinally potent? 
    • What long-term ecological impacts does the wood-intensive cultivation model pose, despite reforestation efforts? 
    • How is Armillaria cultivation managed to prevent pathogen spillover or forest imbalance? 
    • How do pricing trends for Tianma compare across regions? 
    • Is there concern that commodification may impact the retention of traditional knowledge? 
    • Are there ongoing clinical trials in China or globally examining Tianma for the treatment of cardiovascular or neuroprotective diseases? 
    • Please consider comparing CI across genders, age groups, or regions if data are available. 
    • Recent omics-based studies on Tianma are not cited. 
    • Comparative studies on Tianma grown in Zhaotong versus other regions could strengthen claims of superiority. 
    • Include interview questions as supplementary material. 
    • Specify if informants were compensated or what criteria were used for their selection beyond snowball sampling. 
    • Expand the link between biocultural conservation and Sustainable Development Goals. 
    • Reflect on how this case study could serve as a model for other biocultural crops in Asia or globally. 
    • In the Conclusion section, more specific recommendations for policy or industry practice would be beneficial. 
    • Mention potential limitations to enhance transparency.

Author Response

Responses to Reviewer 2:

Comments 1: Have molecular techniques been applied to confirm the identity of Gastrodia varieties or Mycena and Armillaria species?

Response 1: To our knowledge, the Mycena and Armillaria strains used by Zhaotong farmers for Gastrodia elata cultivation are provided jointly by the local government and the Zhaotong Gastrodia Research Institute. These strains have undergone various technical identification procedures by researchers. Relevant information has been presented in the manuscript. Please see Lines 145-154. Furthermore, researchers employed morphological observation, cultivation records, and genomic analysis results to confirm the identity of the Gastrodia varieties. Please see Lines 440-456 for details.

Comments 2: Could folk classifications be supported or contrasted by genetic data?

Response 2: Folk classifications can potentially be supported by genetic data. This perspective is discussed in the manuscript's Discussion section. Please see Lines 440-456 for details.

Comments 3: Were gastrodin and other bioactives quantified across the different forms of elata?

Response 3: We did not perform quantitative analysis ourselves. However, based on literature review, we found that Zhaotong G. elata indeed has higher gastrodin content compared to products from other regions. Interestingly, among Zhaotong variants, bioactive compound levels differ: Yellow Tianma has the highest gastrodin content (14,500 μg/g), followed by Black Tianma (6,000 μg/g), with Green Tianma having the lowest (5,400 μg/g). This appears to contradict the locally held belief that Black Tianma is of the highest quality, warranting further investigation. This point is elaborated in the manuscript. Please see Lines 469-481 for details .

Comments 4: Is there a local perception of which variety is most medicinally potent?

Response 4: Yes, local people generally perceive Wu Tianma as having the strongest medicinal potency. This is stated in the manuscript. Please see Lines 222-224 for details.

Comments 5: What long-term ecological impacts does the wood-intensive cultivation model pose, despite reforestation efforts?

Response 5: This aspect is discussed as a limitation of our study in the Discussion section. Please see Lines 489-500 for details.

Comments 6: How is Armillariacultivation managed to prevent pathogen spillover or forest imbalance?

Response 6: This aspect is discussed as a limitation of our study in the Discussion section. Please see Lines 489-500 for details.

Comments 7:How do pricing trends for Tianma compare across regions?

Response 7: Due to its higher gastrodin content, Zhaotong Tianma commands a higher price compared to products from other regions. This is discussed in the manuscript. Please see Lines 469-472 for details.

Comments 8: Is there concern that commodification may impact the retention of traditional knowledge?

Response 8: This aspect is discussed as a limitation of our study in the Discussion section. Please see Lines 489-500 for details.

Comments 9:Are there ongoing clinical trials in China or globally examining Tianma for the treatment of cardiovascular or neuroprotective diseases?

Response 9: As of June 2025, no directly registered Phase III or higher clinical trials were identified. However, substantial preclinical evidence indicates clear potential for Tianma in cardiovascular and neuroprotective applications, particularly concerning gut-brain axis modulation, antioxidant effects, and anti-inflammatory activity. This is discussed in the manuscript. Please see Lines 463-467 for details.

Comments 10:Please consider comparing CI across genders, age groups, or regions if data are available.

Response 10: We sincerely apologize; based on the current dataset, it is not feasible to compare the Cultural Importance (CI) index across gender, age groups, or regions. However, this is an excellent suggestion. In future research, we plan to consciously collect data to enable such comparisons, which may yield novel insights.

Comments 11: Recent omics-based studies on Tianma are not cited.

Response 11: Following your suggestion, we reviewed recent omics-based studies on Tianma and have cited relevant findings in the manuscript. Please see Lines 445-446 and 457-463 for details.

Comments 12:Comparative studies on Tianma grown in Zhaotong versus other regions could strengthen claims of superiority.

Response 12: Following your suggestion, we have added comparative studies examining Zhaotong Tianma versus products from other regions to strengthen the claims regarding its superior quality. Please see Lines 469-481 for details.

Comments 13: Include interview questions as supplementary material.

Response 13: As requested, the interview questions have been submitted as Supplementary Material 1.

Comments 14:Specify if informants were compensated or what criteria were used for their selection beyond snowball sampling.

Response 14: Following your suggestion, we have rewritten the section detailing informant compensation mechanisms and selection criteria. Please see Lines 155-166 for details.

Comments 15:Expand the link between biocultural conservation and Sustainable Development Goals.

Response 15: Following your suggestion, we have deepened the analysis linking Zhaotong's biocultural conservation efforts to the Sustainable Development Goals (SDGs) in the manuscript. Please see Lines 501-512 and Table 4 for details.

Comments 16: Reflect on how this case study could serve as a model for other biocultural crops in Asia or globally.

Response 16: Following your suggestion, we have discussed the potential of this case study to serve as a model for other biocultural crops in Asia or globally within the Discussion section. Please see Lines 501-512 for details.

Comments 17: In the Conclusion section, more specific recommendations for policy or industry practice would be beneficial.

Response 17: Following your suggestion, we have rewritten the Conclusion section to include more specific recommendations for policy or industry practice. Please see Lines 539-545 for details.

Comments 18: Mention potential limitations to enhance transparency.

Response 18: Following your suggestion, we have dedicated a section in the Discussion to address the study's limitations and shortcomings, enhancing transparency. Please see Lines 489-500 for details.

We sincerely thank the reviewer for their thorough, insightful, and constructive comments, which have been invaluable in improving the depth, clarity, and rigor of our manuscript.

Reviewer 3 Report (New Reviewer)

Comments and Suggestions for Authors

Comments to authors#

Dear authors,

This is a very interesting paper. The results are clearly presented, and the materials and methods have no serious flaws.

Please consider my comments.

General report #

I added a few remarks to the PDF file that is attached.

How is this study different from earlier ethnobotanical research on Gastrodia elata, such as studies on cultivation, symbiotic relationships, or medicinal uses?

Approach: What criteria were used to choose the 114 informants? Were age, gender, and expertise (e.g., farmers vs. healers) balanced to ensure representativeness?

You mention five Gastrodia taxa (one species and four forms) and 23 total species. Are these forms scientifically recognized (e.g., subspecies, cultivars)? Provide references.

Fagaceae Dominance: You note that Fagaceae's slow decomposition supports the growth of Armillaria. Is there empirical data (e.g., decomposition rates, fungal biomass) to substantiate this, or is it inferred from local knowledge?

Cardiovascular Claims: Are the medicinal uses (e.g., treating cardiovascular diseases) supported by pharmacological studies or clinical trials, or are they solely based on traditional knowledge?

Poverty Alleviation: How quantitatively has the Tianma industry impacted poverty reduction (e.g., income data, household surveys)?

Does the current forest conservation policy address the potential overharvesting of Tianma or its symbiotic species (e.g., Armillaria-hosting trees)?

"Xuehong Tianma (unknown form)"—is this a local name for an undescribed taxon? If so, note the need for further taxonomic study.

I have added comments to the tables and figures; please consider them.

Author Response

Responses to Reviewer 3:

Comments 1: How is this study different from earlier ethnobotanical research on Gastrodia elata, such as studies on cultivation, symbiotic relationships, or medicinal uses?

Response 1: Thank you for raising this crucial point regarding our study's novelty. This prompted us to clarify its key distinctions:

(1) Previous research often focused on G. elata's biological traits or singular medicinal value. Our study is the first to systematically examine Zhaotong Tianma within the theoretical framework of "biocultural diversity."

(2) While the "medicine-food homology" attribute has been mentioned previously, systematic investigation was lacking, particularly concerning its integration into local culinary practices. Our work provides the first comprehensive ethnobotanical survey in this genuine producing area, meticulously documenting and analyzing the traditional knowledge system surrounding Tianma, thereby filling the gap in the "food" dimension.

(3) Earlier studies rarely empirically explored the linkage mechanisms between Tianma traditional knowledge practices and regional socio-economic well-being or ecological conservation. Elucidating this dynamic interplay is a core objective of our research.

Comments 2: Approach: What criteria were used to choose the 114 informants? Were age, gender, and expertise (e.g., farmers vs. healers) balanced to ensure representativeness?

Response 2: We apologize for this oversight in the initial manuscript. The selection criteria for the 114 informants have now been detailed in Section 2.2 (Ethnobotanical investigation). Please see Lines 155-166 for the specific procedures addressing age, gender, expertise balance, and representativeness.

Comments 3: You mention five Gastrodia taxa (one species and four forms) and 23 total species. Are these forms scientifically recognized (e.g., subspecies, cultivars)? Provide references.

Response 3: The potential support for folk classifications by genetic data is discussed in the manuscript's Discussion section, along with relevant references. Please see Lines 440-446 for details.

Comments 4: Fagaceae Dominance: You note that Fagaceae's slow decomposition supports the growth of Armillaria. Is there empirical data (e.g., decomposition rates, fungal biomass) to substantiate this, or is it inferred from local knowledge?

Response 4: The conclusion regarding Fagaceae's slow decomposition supporting Armillaria growth is based on empirical knowledge derived from the long-term production practices of local farmers.

Comments 5: Cardiovascular Claims: Are the medicinal uses (e.g., treating cardiovascular diseases) supported by pharmacological studies or clinical trials, or are they solely based on traditional knowledge?

Response 5: The medicinal uses cited, including treating cardiovascular diseases, are based on traditional knowledge and supported by pharmacological studies. This is discussed in the manuscript. Please see Lines 450-467 for details.

Comments 6: Poverty Alleviation: How quantitatively has the Tianma industry impacted poverty reduction (e.g., income data, household surveys)?

Response 6: Following your suggestion, we have supplemented the manuscript with the limited quantitative data on poverty reduction impacts from the Tianma industry that were available during our survey. Please see Lines 397-402 for details.

Comments 7: Does the current forest conservation policy address the potential overharvesting of Tianma or its symbiotic species (e.g., Armillaria-hosting trees)?

Response 7: This specific aspect of policy was not explored within the scope of our current study. We acknowledge this as a limitation and have discussed it in the manuscript's Discussion section (Lines 491-501).

Comments 8: "Xuehong Tianma (unknown form)"—is this a local name for an undescribed taxon? If so, note the need for further taxonomic study.

Response 8: "Xuehong Tianma" is indeed a local name for an undescribed taxon. As suggested, we have clarified in the manuscript (Lines 439-445) that this highlights the need for further taxonomic research.

Comments 9: I have added comments to the tables and figures; please consider them.

Response 9: We have carefully revised the manuscript based on all comments and annotations provided in the PDF file.

Regarding the taxonomic status of G. elata f. flavida, G. elata f. glauca, and G. elata f. viridis: We fully acknowledge that taxonomically, all these forms (f.) are heterotypic synonyms of G. elata. However, to illustrate the significance of folk classification and maintain the article's narrative clarity regarding local knowledge, we retained the distinct forms in our descriptions. To enhance scientific rigor, we explicitly state their synonymy status both within the manuscript (Lines 445-446) and in a footnote to Table 1.

Concerning terminology: "Variety" (var.) is a formal taxonomic rank requiring a Latin diagnosis and type specimen. "Form" (f.) typically denotes minor morphological variations (e.g., flower color). The original naming of these Gastrodia types (f. viridis, etc.) explicitly uses the "f." abbreviation, indicating they were designated as forms, not varieties. Therefore, using "forms" in the manuscript remains the accurate designation for these entities based on their published nomenclature. We trust this interpretation aligns with botanical nomenclature standards.

We are deeply grateful for your meticulous review, insightful questions, and constructive suggestions. Your expertise has significantly strengthened the scientific rigor, clarity, and overall impact of our manuscript.

This manuscript is a resubmission of an earlier submission. The following is a list of the peer review reports and author responses from that submission.

Round 1

Reviewer 1 Report

Comments and Suggestions for Authors

Journal: Biology (ISSN 2079-7737)

Manuscript number: biology-3493183

Manuscript title: Enhancing the Well-being of Indigenous Residents through Biocultural Diversity: A Case Study of Gastrodia elata Blume in Zhaotong City, Yunnan Province, China.

  • In the title “Enhancing the Well-being of Indigenous Residents through Biocultural Diversity: A Case Study of Gastrodia elata Blume in Zhaotong City, Yunnan Province, China” the word “Biocultural” is Inappropriate. The most appropriate word is "bioeconomic" because the issue is related to the economic aspect, which reflects the well-being of the local population, and not the entertainment aspect.
  • L27-28: “However, no scholars have systematically studied the biocultural diversity of Tianma” The researcher must be precise and mention that the plant has not been studied in the study area and not in general because the plant has already been studied previously in other studies.
  • The authors must briefly mention the scientific methodology that the authors followed while conducting the current study in the abstract. and what is related to its novelty and importance in the current manuscript.
  • Plz enrich the abstract with numerical values of study findings.
  • L27, 28, …… and in the entire manuscript; “Tianma” It is preferable for researchers to write the scientific name throughout the manuscript, even if it is abbreviated, and not to write the common name.
  • L44-45: “Keywords” Keywords are needed to be arranged alphabetically. The first letter of keywords should be capitalized. I suggest rephrasing some words because keywords should not repeat words from the title.
  • In part of the introduction, the novelty and importance of this manuscript should be elaborated in detail. Therefore, the main objectives of this work also should be elaborated. The recent related research should be cited focusing on the studied plant.
  • L58: “[10-13]” pls reduce the number of citations in the same place, especially the oldest ones. Pls pay attention to this comment in the entire manuscript.
  • L65: “Gastrodia elata Blume (天麻, Tianma)” plz write the scientific name only
  • Introduction: flow is lost in the middle. Please connect all portions to make a continuous story and reader-friendly.
  • Hypothesis of the work should be very clear at the end of the introduction part.
  • L139: “2.2. Ethnobotanical investigation” this section needs more details to explain the actual work steps throughout the manuscript
  • The experimental method should be described in depth.
  • L155: “2.3. Data analysis” I have major concern on the Data analysis “This part needs more details to know how the data was analyzed statistically to confirm the significance of the results of the manuscript or not.  More detail should be focused on the statistical analysis of the results of the manuscript.
  • Results and discussion should be separated. They are not giving a clear picture in the joint form.  For example, from lines 165-178: This entire part is a discussion of the results and inferences from previous relevant studies. We should have started with the results of the study and then discussed them.
  • Figure 3. The Planting of Tianma in Xiaocaoba Town, Yiliang County, Zhaotong City (A: Germination fungi (Mycena genus) for Tianma seeds; B: Armillaria mellea (Vahl) P. Kumm; C: Baitouma formed from Tianma seeds; D: Asexual reproduction through the planting of Baitouma; E: Tianma’s pollination; F: The utilized fungal wood can also be used as firewood). This format does not add scientific value to the current work, please delete it
  •  Results should be comprehensively explained, which is missing in some parts of the results because of the focus on discussing the results rather than presenting them in detail
  • The Manuscript needs thorough revision to improve the text quality and readability of work.
  • From my point of view, this work is suitable to be developed as a review article rather than a research article in the conventional scientific sense.
  • L250: “our” Personal pronouns should not be written throughout the entire manuscript.
  • Conclusion: is interesting, but it should be very crispy and not in a detailed form.
  • The current study did not provide results that would be published in Biology.
Comments on the Quality of English Language

 The English could be improved to more clearly express the research.

Author Response

Dear Editor:

Thank you very much for the constructive suggestions from you and the reviewers. We have revised our manuscript according to the reviewers’ suggestions. Point-to-point responses to the comments are listed as below. All the co-authors have approved the revised version of this manuscript.

I am very grateful to your considerable assistance with this manuscript. We hope our current efforts have resulted in a revised manuscript that is acceptable for publication in your journal. We look forward to hearing from you. If you have any questions, please do not hesitate to contact me.

Sincerely yours,

Yanxiao Fan 

Yunnan Minzu University

Email: fanyanxiao0510@163.com

Responses to the editor’s and reviewers’ comments:

Reviewer #1:

  • In the title “Enhancing the Well-being of Indigenous Residents through Biocultural Diversity: A Case Study of Gastrodia elataBlume in Zhaotong City, Yunnan Province, China” the word “Biocultural” is Inappropriate. The most appropriate word is "bioeconomic" because the issue is related to the economic aspect, which reflects the well-being of the local population, and not the entertainment aspect.

Thank you for your suggestion, and your comments are well-taken. However, after discussion with the authors, we have decided to retain the term "biocultural diversity" in our article. This is because the article describes the medicinal culture, dietary culture, agricultural culture, and other related aspects surrounding Gastrodia elata Blume (G. elata). Relying on these G. elata-related biocultural elements, the local G. elata economy has achieved sustainable development, which has also improved the quality of life for local residents. Essentially, it is the biocultural aspects associated with G. elata that are playing a crucial role.

  • L27-28: “However, no scholars have systematically studied the biocultural diversity of Tianma” The researcher must be precise and mention that the plant has not been studied in the study area and not in general because the plant has already been studied previously in other studies.

It is indeed a lack of precision in our original statement. We have revised the text to replace the original sentence with "However, no scholar has systematically studied the biocultural diversity of G. elata in Zhaotong." Additionally, we have rewritten the abstract to describe the scientific methods followed in the current research, emphasizing the novelty and significance of the manuscript. For details, please refer to lines 26-51.

  • L27, 28, …… and in the entire manuscript; “Tianma” It is preferable for researchers to write the scientific name throughout the manuscript, even if it is abbreviated, and not to write the common name.

We have replaced the common name of Gastrodia elata with its scientific name throughout the entire manuscript.

  • L44-45: “Keywords” Keywords are needed to be arranged alphabetically. The first letter of keywords should be capitalized. I suggest rephrasing some words because keywords should not repeat words from the title.In part of the introduction, the novelty and importance of this manuscript should be elaborated in detail. Therefore, the main objectives of this work also should be elaborated. The recent related research should be cited focusing on the studied plant.

Based on the suggestions received, we have revised the keywords and elaborated on the novelty and importance of this manuscript in the introduction section, highlighting the main objectives of this work. For specific details, please refer to lines 52-53 and 103-117.

  • L58: “[10-13]” pls reduce the number of citations in the same place, especially the oldest ones. Pls pay attention to this comment in the entire manuscript.

According to the requirements, we have made adjustments to the references in the entire manuscript.

  • L65: “Gastrodia elata Blume (天麻, Tianma)” plz write the scientific name only. Introduction: flow is lost in the middle. Please connect all portions to make a continuous story and reader-friendly.Hypothesis of the work should be very clear at the end of the introduction part.

We have replaced the common name of G. elata with its scientific name throughout the entire manuscript. Additionally, we have adjusted the introduction section to make it more coherent and have emphasized the main objectives of this work at the end of this section. For specific details, please refer to lines 103-117.

  • L139: “2.2. Ethnobotanical investigation” this section needs more details to explain the actual work steps throughout the manuscript. The experimental method should be described in depth.

As requested, we have added detailed information about the investigation process in the "2.2. Ethnobotanical investigation" section. Please refer to lines 160-168 for specifics.

  • L155: “2.3. Data analysis” I have major concern on the Data analysis “This part needs more details to know how the data was analyzed statistically to confirm the significance of the results of the manuscript or not. More detail should be focused on the statistical analysis of the results of the manuscript.

In accordance with the requirements, we have supplemented the content related to data analysis. In this study, we used the coefficient of determination (R²) to analyze the relevant data and constructed a simple linear regression model to support the observed trends, explaining the relationship between the cultivated area of G. elata and comprehensive economic growth. Please refer to lines 177-194 and 431-435, as well as Figure 6D, for specifics.

  • Results and discussion should be separated. They are not giving a clear picture in the joint form.  For example, from lines 165-178: This entire part is a discussion of the results and inferences from previous relevant studies. We should have started with the results of the study and then discussed them.

Thank you for your suggestions, which have been very helpful in improving the quality of this manuscript. We have deleted many unnecessary contents in the results and discussion sections to present the research findings more intuitively.

  • Figure 3. The Planting of Tianma in Xiaocaoba Town, Yiliang County, Zhaotong City (A: Germination fungi (Mycena genus) for Tianma seeds; B: Armillaria mellea(Vahl) P. Kumm; C: Baitouma formed from Tianma seeds; D: Asexual reproduction through the planting of Baitouma; E: Tianma’s pollination; F: The utilized fungal wood can also be used as firewood). This format does not add scientific value to the current work, please delete it.

Based on your suggestion, we have removed Figure 3 from the manuscript.

  • Results should be comprehensively explained, which is missing in some parts of the results because of the focus on discussing the results rather than presenting them in detail. The Manuscript needs thorough revision to improve the text quality and readability of work.From my point of view, this work is suitable to be developed as a review article rather than a research article in the conventional scientific sense.

Based on your suggestion, we have revised the entire manuscript, with a focus on the results section, to enhance the text quality and readability of the work.

  • L250: “our” Personal pronouns should not be written throughout the entire manuscript.Conclusion: is interesting, but it should be very crispy and not in a detailed form.

In accordance with your suggestions, we have removed personal pronouns such as "our" from the manuscript and have made revisions and adjustments to the conclusion section.

  • The English could be improved to more clearly express the research.

Thank you for your suggestion. We have once again had the manuscript edited by a native English speaker to improve the language.

Reviewer 2 Report

Comments and Suggestions for Authors
  1. The authors claim that local people classify Gastrodia elata into four types; however, this folk classification lacks molecular or phytochemical validation.
  2. Although elata is classified as vulnerable by the IUCN and listed in CITES Appendix II, the manuscript fails to address population trends or conservation efforts beyond local forest protection. The authors should include data on wild populations, elaborate on conservation challenges, and provide insights into how G. elata populations are monitored.
  3. The study mentions the medicinal benefits of elata, such as treating cardiovascular diseases, but lacks citations for clinical or pharmacological studies validating these effects. If available, the authors should provide evidence from clinical trials or in vivo studies.
  4. The study presents data on Tianma cultivation area, economic growth, and forest coverage; however, it lacks statistical analyses to support the observed trends.
  5. The snowball sampling method used to select the 114 participants introduces bias, as it relies on pre-existing networks rather than random sampling.
  6. While the study claims to use ethnobotanical methods, it lacks a detailed explanation of data validation and reliability assessment.
  7. The authors should clarify how they distinguish the economic impact of elata from other local industries and describe other economic activities that contribute to poverty alleviation in Zhaotong.
  8. The Figure 1 legend should be expanded to provide a more detailed description of the map and its components. Specifically, what the map shows, locations, and symbols used.
  9. The Figure 2 legend should specify which parts of Gastrodia elata are depicted in the figure (rhizome/ flower/ whole plant).
  10. Add a note at the bottom of each table explaining the meaning of the "/" symbol.
  11. The authors should expand the discussion of the study's limitations, provide more detailed future research directions, and discuss the potential outcomes and implications of their findings.

Comments on the Quality of English Language

 The English could be improved to express the research more clearly.

Author Response

Dear Editor:

Thank you very much for the constructive suggestions from you and the reviewers. We have revised our manuscript according to the reviewers’ suggestions. Point-to-point responses to the comments are listed as below. All the co-authors have approved the revised version of this manuscript.

I am very grateful to your considerable assistance with this manuscript. We hope our current efforts have resulted in a revised manuscript that is acceptable for publication in your journal. We look forward to hearing from you. If you have any questions, please do not hesitate to contact me.

Sincerely yours,

Yanxiao Fan 

Yunnan Minzu University

Email: fanyanxiao0510@163.com

Responses to the editor’s and reviewers’ comments:

Reviewer #2:

  • The authors claim that local people classifyGastrodia elata into four types; however, this folk classification lacks molecular or phytochemical validation.

Thank you to the reviewer for the valuable comments. We fully understand your concern regarding the lack of molecular or phytochemical validation for the ethnobotanical classifications. Indeed, the four Gastrodia elata types mentioned in this paper are based on local traditional knowledge and the ethnobotanical classification system, which have been widely accepted and used to distinguish different varieties of G. elata in long-term practice. We believe that this classification method has important reference value, especially in describing the morphological characteristics, growth habits, and traditional uses of Gastrodia elata. Studies by others have also confirmed the scientific validity of this ethnobotanical classification, as detailed in lines 205-215.

  • Although elata is classified as vulnerable by the IUCN and listed in CITES Appendix II, the manuscript fails to address population trends or conservation efforts beyond local forest protection. The authors should include data on wild populations, elaborate on conservation challenges, and provide insights into how G. elata populations are monitored.

Thank you for your suggestion. Since the focus of this manuscript is on the biocultural diversity of G. elata, and the majority of the G. elata utilized in Zhaotong were cultivated, we did not pay excessive attention to the conservation of wild G. elata populations. However, your suggestion is indeed meaningful. In our upcoming work, we will incorporate the topics of population trends or conservation efforts beyond local forest protection, as well as wild populations and a detailed elaboration on conservation challenges, into our future research plans. Nevertheless, to avoid any potential misunderstandings, we have removed the discussion regarding IUCN and CITES from the text to ensure the article flows more coherently.

  • The study mentions the medicinal benefits of elata, such as treating cardiovascular diseases, but lacks citations for clinical or pharmacological studies validating these effects. If available, the authors should provide evidence from clinical trials or in vivo studies.

As requested, we have provided clinical and pharmacological evidence for the pharmacological activities of G. elata in the manuscript, specifically in lines 314-327.

  • The study presents data on Tianma cultivation area, economic growth, and forest coverage; however, it lacks statistical analyses to support the observed trends.

In accordance with the recommendations, this study utilizes the coefficient of determination (R²) to analyze the relevant data and establishes a simple linear regression model to substantiate the observed trends, thereby interpreting the relationship between the cultivation area of G. elata and comprehensive economic growth. For detailed information, please see lines 177-194, 431-435, and Figure 6D.

  • The snowball sampling method used to select the 114 participants introduces bias, as it relies on pre-existing networks rather than random sampling.

We sincerely appreciate your valuable feedback. We fully acknowledge that the snowball sampling method may introduce bias, as it relies on pre-existing networks rather than strict random sampling. However, we chose this approach primarily because our target population consists of individuals with traditional knowledge of Tianma, a group that is highly specific and difficult to access through random sampling. Snowball sampling allowed us to more effectively reach these individuals. That said, we are aware of the limitations of this method. To mitigate potential biases, we established clear inclusion criteria to ensure diversity in key characteristics (such as age, gender, geographic location, etc.) within our sample.

  • While the study claims to use ethnobotanical methods, it lacks a detailed explanation of data validation and reliability assessment.

We would like to extend our gratitude to the reviewers for their valuable comments. We fully agree that data validation and reliability assessment are critical components of ethnobotanical research. In our study, we employed multiple data verification methods: (1) Cross-referencing information from various sources (such as literature records, field survey data, and expert interviews) to ensure consistency and accuracy. (2) During field surveys, we invited multiple local informants (e.g., traditional healers, farmers, or elders) to independently verify the same information, thereby reducing the impact of individual biases. (3) For specific uses of G. elata, we also validated the information provided by local informants through on-site observations and documentation. Additionally, in accordance with the recommendations, this study utilizes the coefficient of determination (R²) to analyze the relevant data and establishes a simple linear regression model to substantiate the observed trends, thereby interpreting the relationship between the cultivation area of G. elata and comprehensive economic growth. For detailed information, please see lines 177-194, 431-435, and Figure 6D.

  • The authors should clarify how they distinguish the economic impact of elata from other local industries and describe other economic activities that contribute to poverty alleviation in Zhaotong.

Thank you for your suggestion. As this study primarily focuses on G. elata, the investigation did not extensively address the impact of economic growth on other local industries or other economic activities contributing to poverty alleviation in Zhaotong. However, we recognize the significant role that other economic activities play in driving local development. Therefore, in future research, we aim to further explore the synergistic effects between the G. elata industry and other economic activities, as well as how multi-industry collaboration can achieve more comprehensive economic development and poverty alleviation goals.

  • The Figure 1 legend should be expanded to provide a more detailed description of the map and its components. Specifically, what the map shows, locations, and symbols used.

As requested, we have refined Figure 1 by adding detailed explanations to the legend.

  • The Figure 2 legend should specify which parts of Gastrodia elata are depicted in the figure (rhizome/ flower/ whole plant).

In accordance with the requirements, we have enhanced Figure 2 by adding annotations to various parts of the image.

  • Add a note at the bottom of each table explaining the meaning of the "/" symbol.

Following the recommendation, we have added an explanation of "/" at the bottom of the table.

  • The authors should expand the discussion of the study's limitations, provide more detailed future research directions, and discuss the potential outcomes and implications of their findings.

As requested, we have revised the conclusion section of the study. In this section, we have expanded the discussion on the limitations of the research, provided more detailed future research directions, and discussed the potential implications and significance of the findings. For specifics, please refer to lines 510-539.

Round 2

Reviewer 1 Report

Comments and Suggestions for Authors

Dear Editor

I am still not satisfied with the corrections made to the manuscript, as it has not improved enough to be published in Biology. The manuscript has not been adequately commented on.

Comments on the Quality of English Language

The English could be improved to express the research more clearly.